

# A new 1500-year-long varve thickness record from Labrador, Canada, uncovers significant insights into large-scale climate variability in the Atlantic

François Lapointe[1*], Antoine Gagnon-Poiré[2,3,4,a*], Pierre Francus[2,3], Patrick Lajeunesse[5], Clarence Gagnon[2,b]

[1]Climate System Research Center, Department of Earth, Geographic, and Climate Sciences, University of Massachusetts, Amherst, MA 01003, USA

[2]Centre Eau Terre Environnement, Institut national de la recherche scientifique, Québec, QC G1K 9A9, Canada

[3]GEOTOP, Research Centre on the Dynamics of the Earth System, Montréal, QC H2X 3Y7, Canada

[4]Centre d'études nordiques, Québec, QC G1V 0A6, Canada

[5]Département de géographie, Université Laval, Québec, QC G1V 0A6, Canada

[a]Antoine Gagnon-Poiré is currently at the Ministère de l'Environnement, de la Lutte contre les changements climatiques, de la Faune et des Parcs, Québec, Canada

[b]Clarence Gagnon is currently at the Département de génie civil et de génie des eaux, Université Laval, Québec, QC G1V 0A6, Canada

*These authors contributed equally to the work

Correspondence to: Pierre Francus (pierre.francus@inrs.ca)

**Abstract.** Grand Lake, located in Labrador, at the northeastern margin of North America, is a deep lacustrine basin that contains a well-preserved annual laminations record spanning the interval 493 to 2016 CE (1524 years). The chronology of this new varved sequence is established from layer counting of high-resolution images of thin sections. Radiometric dating ($^{137}$Cs and $^{14}$C) validates the reliability of the varve chronology. Varve thickness is significantly correlated ($r = 0.38$) with the total precipitation recorded at the nearest weather station Goose A. The varve thickness series reveals high values during the 1050–1225 CE period, that is corresponding to the Medieval Climate Anomaly, whereas the 15th–19th centuries, related to the Little Ice Age, shows low values. The teleconnections between several Goose A instrumental data series and some modes of climate variability such as the winter Greenland Blocking (negative North-Atlantic Oscillation) and the significant correlations between our varve thickness record and three other Northern Hemisphere high-resolution proxy records suggest that the Grand Lake record tracks North-Western Atlantic large-scale mode of hydroclimate variability over the past ~1500 years.

## 1 Introduction

Northeastern Canada experiences significant interannual to multidecadal climate variability driven by large-scale atmospheric and oceanic patterns, such as the North Atlantic Oscillation (NAO) and the Atlantic Multidecadal Variability (AMV) (Banfield and Jacobs, 1998; Boucher et al., 2017; Chartrand and Pausata, 2020; D'Arrigo et al., 2003; Dinis et al., 2019; Durkalec et al.,



2016; Finnis and Bell, 2015; Way and Viau, 2015). This makes the region crucial for studying the Western North Atlantic

hydroclimate system. However, the hydrological response to multidecadal climate variability remains poorly understood in northern Atlantic regions (Linderholm et al., 2018; Ljungqvist et al., 2016; IPCC 2013). This gap is particularly evident in eastern Canada, where only two annually resolved hydrological reconstructions of boreal catchments exist, covering the last two centuries: one based on tree-ring datasets (Boucher et al., 2011; Nasri et al., 2020; Dinis et al., 2019; Nicault et al., 2014), and another derived from a short varve sediment sequence from Grand Lake (Gagnon-Poiré et al., 2021). Given that

hydroelectricity is the primary energy source in eastern Canada, understanding the long-term evolution and mechanisms influencing hydroclimatic regimes is essential for sustainable planning. The short varve record at Grand Lake covering the 1856–2016 period demonstrated great potential for hydrological reconstruction (Gagnon-Poiré et al., 2021). Longer proxy-based reconstructions are still required for improving our knowledge on long-term regional hydrological variability. Developing longer annually resolved palaeohydrological records remains, however, a challenge due to the lifespan of trees

which rarely exceeds 300 years and because lakes containing well defined and continuous long annually laminated sequences are rare in boreal regions (Ramish et al., 2020).

In this study, we present a new varve record from Grand Lake, Labrador that spans the last 1500 years. A new core located in the distal part of the sedimentary basin allows extending the Grand Lake varve dataset to the millennium scale. By using this new long Grand Lake varved sequence, this paper aims at producing the first reconstruction with annual resolution covering

the last fifteen centuries at the western fringe of the Atlantic Ocean, allowing to improve our knowledge of the Western North Atlantic large-scale mode of climate variability.

## 2 Regional setting

Grand Lake, Labrador, is a 245 m-deep and 55 km-long fjord lake (Fig. 1) deglaciated ca. 8000 years ago (Dalton et al., 2020; Fulton and Hodgson, 1979; Occhietti et al., 2011; Trottier et al., 2020). The lake is located at the eastern margin of North

America in the high boreal forest ecoregion, one of the most temperate climates in Labrador. This region is influenced by temperate continental westerly and southwesterly winds and maritime conditions from the Labrador Sea and Labrador Current. Winter temperatures can fall well below freezing, with average lows between -10°C and -20°C. In contrast, summers are cool, with average temperatures ranging from 10°C to 15°C. Yet, the region experiences strong seasonal variability, with long, harsh winters and short, temperate summers. Labrador's climate is moist, receiving ample annual precipitation, primarily from snow

during the long winter months and rain in the summer. Annual precipitation totals are between 800 and 1200 mm, depending on proximity to the coast, with the heaviest amounts near the Labrador Sea due to maritime influences. Freeze-up of local lakes occurs in late fall, and ice break-up happens in late spring to early summer. Ice coverage on the lake is significant due to the cold air and water masses associated with the Labrador Current, which prolongs the winter-like conditions and result in





snow cover persisting from late October to early May. Hence, sediment deposition can occur only from early May to late
November through snow melt and summer rain events.

The lake is fed by two large rivers located at its western end (Fig. 1b) that transport a substantial amount of sediments. The
Naskaupi River supplies ~70 km$^3$ a$^{-1}$ of freshwater (Kamula et al., 2017) into Grand Lake and is the second largest river in
Labrador. The Beaver River is the second main tributary of the lake. Regional streamflow regime is classified as nival
(snowmelt dominated) (Bonsal et al., 2019). Grand Lake flows into a small tidal lake (Little Lake) and subsequently towards
Lake Melville by the small town of North West River.

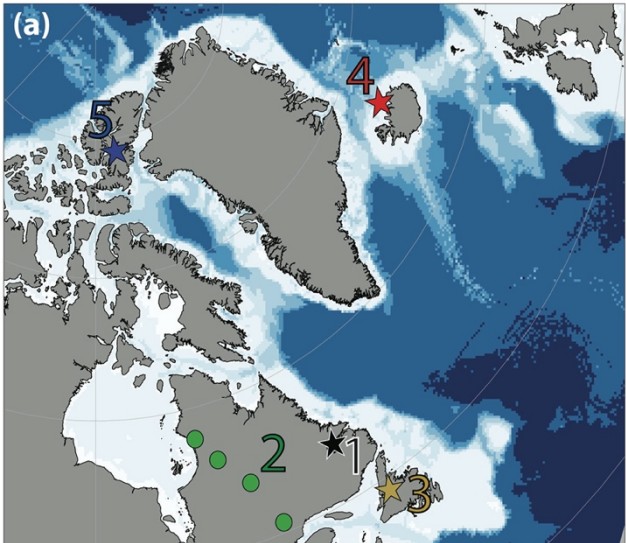

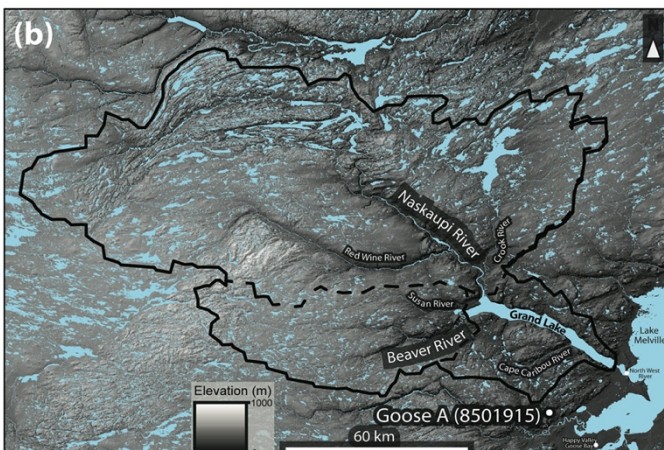

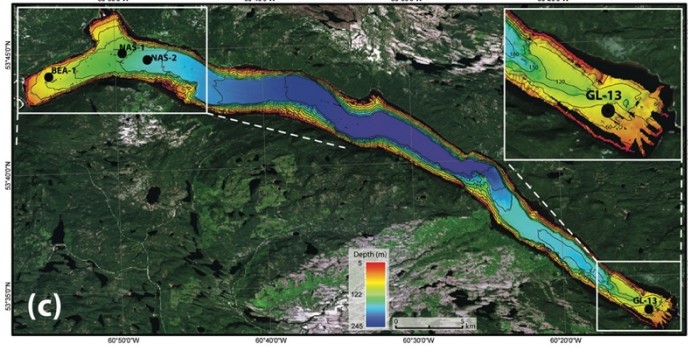

**Figure 1: (a) Location map of the site and the records mentioned in the paper. (b) Grand Lake watershed (black line) and its main tributaries. The Goose A weather station (8501915) is represented by the white dot. (c) High-resolution swath bathymetry (3 m resolution) of Grand Lake (Trottier et al., 2020) coupled with a Landsat image (USGS) and coring site locations Isobath (30 m)**
**Cores BEA-1 NAS-1 et NAS-2 are short proximal cores from Gagnon-Poiré et al. (2021); GL-13 points to the location of the cores from this study. Modified from Gagnon-Poiré et al. (2021).**





## 3. Methods

### 3.1 Sediment coring and analysis

Sediment cores were collected at four sites in Grand Lake (Fig. 1c). During winter 2017, cores BEA-1, NAS-1 and NAS-2 (30
to 120 cm-long) were collected near Grand Lake's main tributaries using a UWITEC percussion corer (Gagnon-Poiré et al., 2021) at depths of 90 to 165 m. Proximal sites BEA-1, NAS-1 and NAS-2 and their 160 year-long (1856-2016) varve sequences were previously presented and discussed in Gagnon-Poiré et al. (2021). These cores were used to reconstruct the average discharge (*Q*-mean) of the Naskaupi river.

Core GL17-13A (320 cm-long) was collected in 2017 using a Rossfelder Corp. submersible vibracorer at a site near the lake
outlet at a depth of 75 m, ~45 km downstream from the lake head (Fig. 1c). This site has a lower sedimentation rate allowing for reaching older sediments. The aluminum core tube was cut in the field in three sections to ease its transport. Since the upper 10 cm of the vibracore GL17-13A were not well preserved, two other cores (GL20-13A and GL20-13B) were collected during winter 2020 from the same site using a UWITEC percussion corer to obtain a better quality upper sedimentary sequence. The cores were first scanned using a Siemens SOMATOM Definition AS+ 128 medical CT-Scanner for previsualization and
identification of laminated facies. The cores were then opened, described and photographed with a high-resolution line-scan camera mounted on an ITRAX core scanner (RGB colour images; 50 μm-pixel size). Geochemical X-ray Fluorescence (μXRF) analysis was performed on the core half (30 kV and 30 mA) using the same instrument at a down-core resolution of 100 μm and an exposure time of 5 seconds. μXRF element profiles were used to visualize varves boundaries and their sub-layer facies (Croudace et al., 2006; Cuven et al., 2010; Kylander et al., 2011).
Thirty-seven overlapping thin sections were made to cover the entire GL-13 sedimentary sequence following Normandeau et al. (2019). Digital images of the thin sections were obtained using a transparency flatbed scanner at 2400 dpi resolution (1 pixel = 10.6 μm) using two polarized filters at 90°, sandwiching the thin section between the filters, to produce crossed-polarized light. Thin sections were also observed using a Zeiss EVO 50 Scanning Electron Microscope (SEM) in backscattered mode using the custom-made *Analyse Image* software (Francus and Nobert, 2007) to obtain a continuous profile of high-
resolution images (pixel size = 1 μm) (Lapointe et al., 2012).

### 3.2 Chronology

### 3.2.1 Counting and measuring laminations

A composite laminated sediment sequences was assembled from the overlapping thin sections, starting with GL20-13B that better preserved surface sediments (Fig. 2a). Correlation between the two cores was made using markers beds clearly visible
in both cores. The laminations were counted twice on the crossed-polarized images (pixel size = 10.6 μm) of the thin sections by two independent observers (Antoine Gagnon-Poiré (AGP) and Clarence Gagnon (CG)) using *Analyse Image* software (Francus and Nobert, 2007). The counts and the position of the varve boundaries were validated using SEM images in backscattered mode (Fig. 2d) that allow a better view of the sediment structure thanks to a strong contrast between the matrix



and the grains (Francus 1998). On areas where the two initial counts were not similar, two complementary counts were made
using the *PeakCounter* software (Marshall et al., 2012) that uses the µXRF profiles as additional information. An error estimate
was calculated based on the difference in the number of laminations counted.

### 3.2.2 Radiometric dating

Two wood fragments located on CT scan radial images were handpicked from core GL17-13A for radiocarbon dating. Samples
were sent to the Radiochronology Laboratory of the Centre d'Études Nordiques (Université Laval, Québec) for a HCl-NaOH-
HCl pre-treatment and graphitization. The dating was performed by accelerator mass spectrometry (AMS) at the Earth System
Science Department Keck Carbon Cycle AMS Facility at the University of California at Irvine. The dates obtained were
calibrated with Calib 7.1 (Stuiver et al., 2018) using the IntCal20 database (Reimer et al., 2020) and are shown in Table 1.
Age-depth models based on radiocarbon dating was performed with the ClamR software version 2.2 (Blaauw 2010) using a
linear interpolation for the site GL-13 with 95% confidence interval.

**Table 1 AMS $^{14}$C age with 2 sigma of the dated material from the site GL-13**

| Core name | Depth (cm) | Material | Laboratory number | Conventional $^{14}$C age BP ($\pm 2\sigma$) | Calibrated age CE (median probability) |
|---|---|---|---|---|---|
| GL17-13A-V | 192 | Wood fragment | UCIAMS-205590 | 1560 ± 15 | 492 (434-565) |
| GL17-13A-V | 214 | Wood fragment | UCIAMS-205582 | 1710 ± 15 | 359 (256-406) |

Two-cm thick subsamples from the GL20-13B surface sediments cores were measured for Cesium-137 ($^{137}$Cs) activity (Ritchie
and McHenry, 1990) using a high-resolution germanium diode gamma detector and multichannel analyzer gamma counter.
The $^{137}$Cs activity was used to identify layers deposited during the 1963–1964 peak of nuclear tests.

### 3.3 Spatial correlation

Correlation maps of instrumental climate data were prepared using the Climate Explorer tool that is managed by the Royal
Netherlands Meteorological Institute (van Oldenborgh and Burgers, 2005). Atmospheric pressure data are from ERA-Interim
reanalysis (Dee et al. 2011), sea surface temperature data are the Extended SST v5 from NOAA (National Oceanic and
Atmospheric Administration) (Huang et al., 2017), and precipitation anomalies from the NCEP/NCAR (National Centers for
Environmental Prediction/National Center for Atmospheric Research Reanalysis) (Kalnay et al., 1996).



## 4 Results

### 4.1 Sediment facies

Sediments retrieved at site GL-13 near the lake outlet consist of laminated minerogenic material (Fig. 2). The upper section of core (0-200 cm) comprises clear and distinct thin laminations having an average thickness of 1.26 mm (Fig. 2b, c). The observed laminations are horizontal and continuous over this entire interval. The laminations are made of 2 layers. The base layer is a silty-clay sediment matrix containing angular and rounded grains ranging from very fine sands to fine silts (Fig. 2c, d). The upper layer of the lamination consists of a clay cap rich in Fe (Fig 2 c, d).

Below 200 cm depth, these thin laminations disappear marking a clear stratigraphic transition.





**Figure 2: (a) Photograph of core GL17-13A and GL20-13B overlain by thin section images constituting the composite sequence. Black horizontal lines indicate the core cuts. (b) Photograph of a thin section showing the distal varve facies from site GL-13. Thin sections are overlain by iron relative intensities obtained using μXRF (yellow line) and by horizontal white bars marking the varve boundaries. (c) Close-up of a thin-section scan, with the varve boundaries marked by the horizontal yellow bars and the location of SEM images marked by yellow boxes with their red ID. (d) Backscattered image of a varve, with the vertical white boxes showing the varve extent.**



## 4.2 Chronology

Core GL20-13B was used to build the upper part of the chronology as it was not disturbed. The first complete lamination
below the sediment surface was considered to represent CE 2017 as two additional laminations were visible in the parallel
disturbed core (i.e., GL20-13A). The upper laminated sequence chronology is consistent with the Cesium-137 main peaks
found at 5 cm depth in core GL20-13B (Fig. 3, Fig. S1), which confirm the varved nature of the distal laminations. Varve
counts made between thick (coarse) distinctive laminations present in both cores GL17-13A and GL20-13B were identical,
providing confidence that a composite sequence can be established (Fig. S1). The composite sequence count starts at the
surface of the core GL20-13B (2017 CE) and switches to core GL17-13A at marker bed dated 1794 CE in the varve chronology
(Fig. 2a, Fig. S1). Three years was added at each of the two cuts made on the core GL17-13A. A 5.6 mm thick non-erosive
rapidly deposited layer (RDL), dated CE 1392 was removed from the sequence.

For the composite core GL-13, the two counts using *Analyse Image* software show a very low counting error (± 0.13%), with
a difference of only 4 laminations between them (Fig. 3). Sediment accumulation rates (or age depth model, Fig. 3) indicate
that the input of clastic sediment at this site was relatively steady throughout the past 1500 years.

The base of the varve chronology in core GL-13 at 190.9 cm depth is 594 CE, in continuity with 2 radiocarbon dates (Table
1) that are 492 CE (192 cm depth) and 359 CE (214 cm depth).

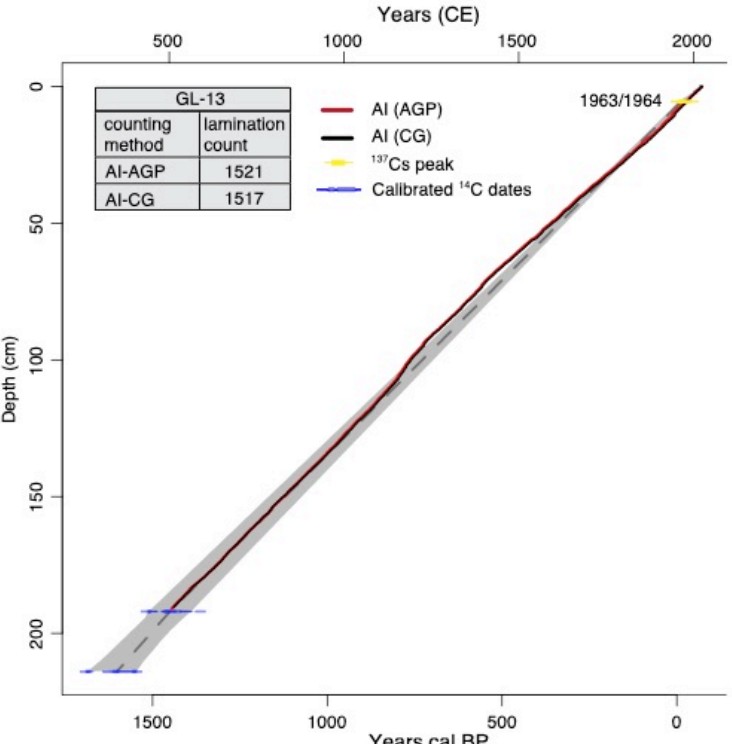

**Figure 3: Comparison of age-depth models for sites GL-13 based on two individual lamination counts (A.G.P. and C.G.) from thin**
**section using Image Analysis (IA) and from $^{137}$Cs and AMS $^{14}$C dating.**




### 4.3 The varve thickness record

The raw varve lamination thickness time series of cores GL-13 is presented in Fig. 4. A log normal time-series was also produced for comparisons with other records (Fig. 6, 7). Varves are thicker than the average between 1050 and 1225 CE. This interval corresponds to Medieval Climate Anomaly (950 to 1250 CE as defined by Mann et al. 2009), with the thickest varves occurring 1164, 1383, 1215 1143, 1141 and 1132 CE. Varves are persistently thinner than average during the 1400–1875 CE interval, an interval a little longer and belated than the Little Ice Age (~1400 – 1800 CE, Mann et al. 2009). The recent decrease of varve thickness (Fig. 5) is within the limits of the variability of the last 1500 years. There is pronounced multidecadal variability throughout much of the record, particularly noticeable between ~1100 and 1225 CE, as well as from 1500 to 1800 CE (Fig. S2).

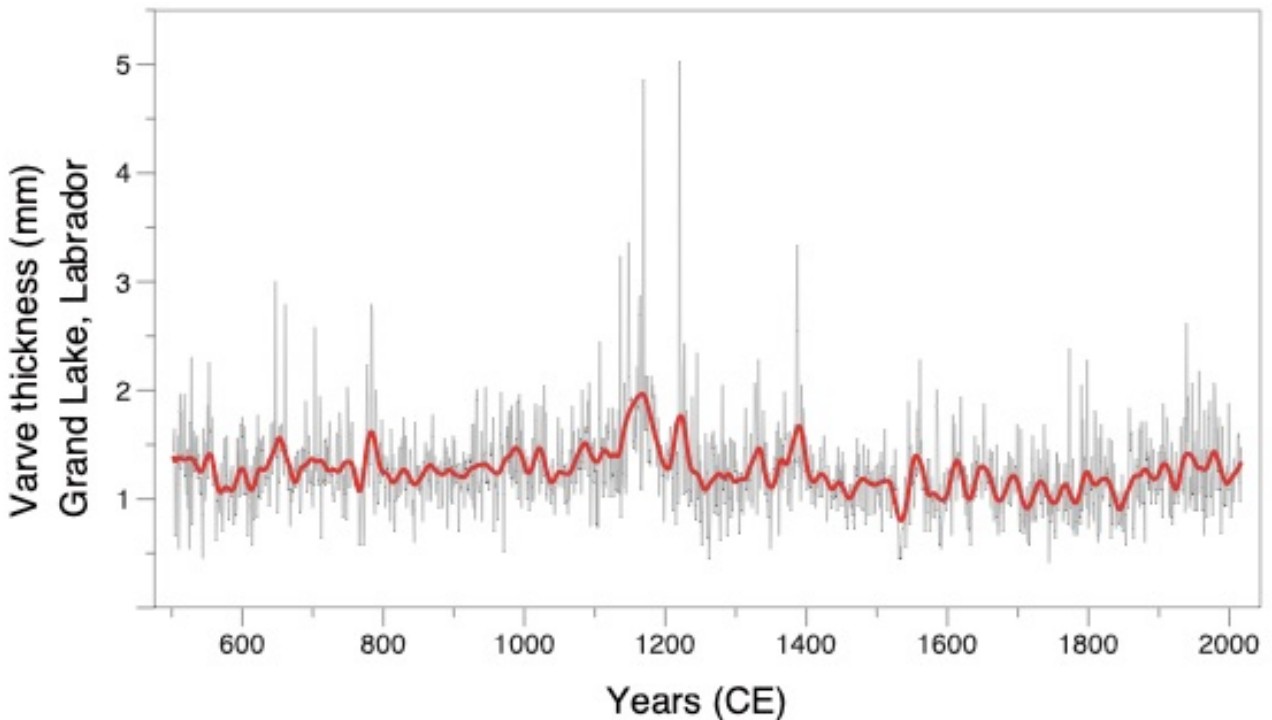


**Figure 4: Varve thickness time series from the composite core of site GL-13 with a 30-year loess first order low-pass filter (red line).**

### 4.4 Varve thickness based time-series and local instrumental records

We now investigate the relationship between the longest instrumental record in the region, provided by the Goose A weather station (Fig. 1b), and both the proximal NAS site and the newly developed distal varve thickness (VT) record GL17-13A (Fig. 1c). The analysis reveals significant positive correlations between both the proximal and distal VT sites and the total precipitation recorded at Goose A, with correlation coefficients of $r = 0.39$ ($p<0.001$) and $r = 0.38$ ($p<0.001$), respectively (Fig. 5a, b). A significant correlation is also observed between distal VT and snow precipitation at Goose A ($r = 0.31$, $p =$





0.006) (Fig. 5c), whereas no significant correlation is found between VT and rainfall amounts at this distal site ($r = 0.15$, $p =$ 0.19). These results suggest that snow precipitation is the primary driver of the VT record. It can be also seen that both datasets

(VT and precipitations) exhibit a marked decline in precipitation signals around the late 1980s, with a particularly pronounced reduction in snow precipitation during this period.

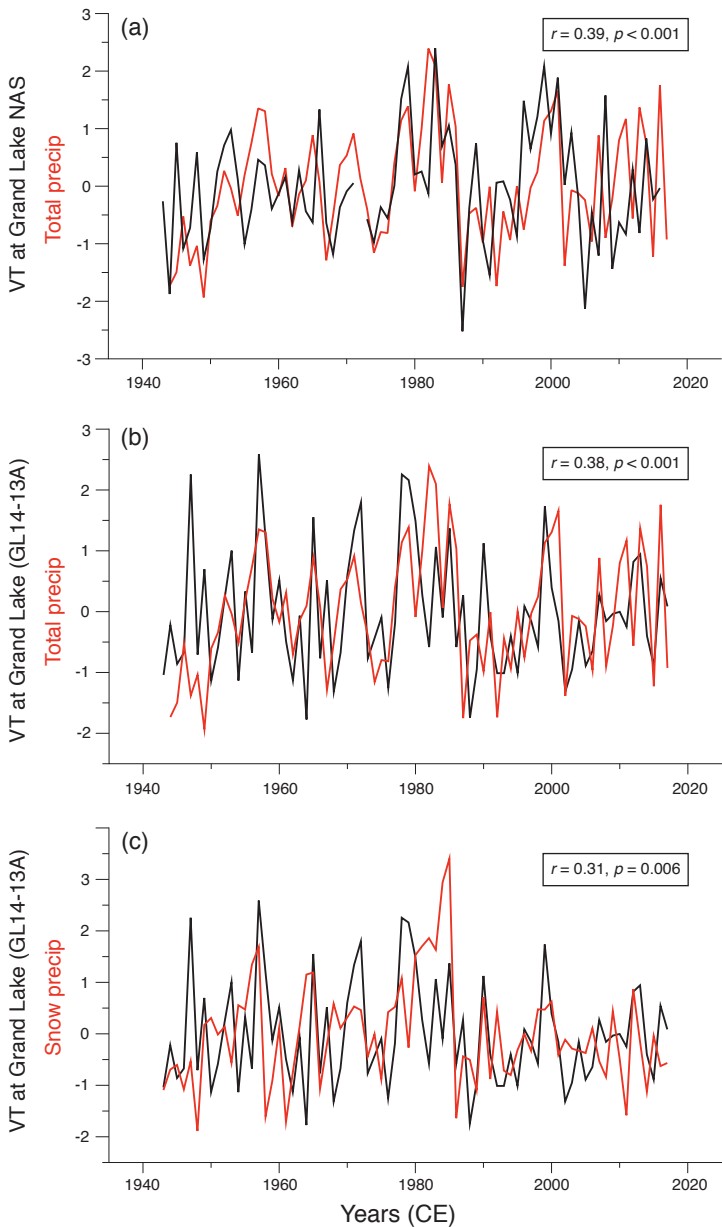

**Figure 5: Comparison between Goose A precipitation record and varve thickness. (a) Total precipitation at Goose A and VT at NAS. (b) Total precipitation at Goose A and VT at GL17-13A. (c) Snow precipitation at Goose A and VT at GL17-13A.**



## 5 Discussion

### 5.1 Varve formation

The lamination in the distal site GL-13 are clastic varves following the classification of Zolitschka *et al.* (2015). The fine component of the basal layer is interpreted as the settling of the very distal fluvial plume flowing from the Naskaupi and Beaver Rivers floods that is travelling across the lake as overflow. The continuousness of these hypopycnal flows across Grand Lake is favoured by the water column stratification (Gagnon-Poiré, 2023). These currents seem capable of transporting large quantities of sediment as evidenced by high sediment accumulation rates at the mouth of the outlet of the lake, i.e., the North-West River (Kamula et al., 2017), and by satellite images of Grand Lake showing plumes reaching the distal site GL-13 during spring discharges (Fig. S3). Other potential sources of sediment are *Rapidly Deposited Layers* (RDL) that are commonly found in deep fjord lakes (St-Onge et al., 2004). The RDLs are sporadic events that can yield large amount of sediment in the water column from many potential locations all around the lake, but are usually coarse-grained and expressed as hyperpycnal flows. Three trigger mechanisms could be responsible for the RDL deposition (St-Onge et al. 2004): floods, landslides initiated by overloading or oversteepening, and earthquakes. The latter are unlikely since the closest seismic zone is the Saguenay fjord area that is 900 km away. Only one RDL was found in our distal sequence most probably because the coring site is located on a relatively shallow location in the lake that is less likely reached by hyperpycnal flows. The isolated silt and sand grains (Fig. 2d) are thought to be ice-rafted for the angular ones, and, for the rounded ones, sourced from one of the RDL or of the several minor streams proximal to the coring site. The clay cap is forming in winter under lake ice as it is the case for all clastic varves and constitute main criteria for identifying a year of sedimentation (Zolitschka *et al.* 2015).

### 5.2 Hydroclimate influence on varve properties

Gagnon-Poiré et al. (2021) already demonstrated that Grand Lake proximal varves are strongly linked with spring discharge conditions: varve thickness and particle size measurements show significant positive correlation with observed Naskaupi River nival runoff (April to July) ($r = 0.58$, $p = {<}0.0001$). The discharge record is strongly influenced by the amount of snow accumulating during the cold season in this region when temperature rise above zero in April (Fig. S4). Snowmelt is not the only factor controlling river discharge: rain also becomes a significant contributor to precipitation in April, with relatively high values in May and June, which coincides with the maximum discharge of the Naskaupi River (Fig. S5). This combined influence is illustrated by the correlation between the total precipitation recorded at Goose A and the VT records at the proximal site (Fig. 5a) and at the distal site (Fig. 5b) ($r = 0.38$, $p = {<}0.001$).

### 5.3 Grand Lake varve record comparison with regional proxy data

Our new 1500-year varve thickness (VT) record from GL provides a valuable opportunity to explore regional long-term hydroclimatic changes by comparing it with other high-resolution records. The log-normal transformed VT series is correlated to the maximum latewood density (MXD) from a tree network in Eastern Canada over the past ~1200 years (Wang et al. 2023,





Fig. 6a). MXD is a proxy for summer (May to August) temperature, suggesting that temperature influences the VT record at GL. A highly resolved sea surface temperature (SST) reconstruction off North Iceland, based on alkenones with a four-year resolution (Sicre et al. 2008), also aligns with the GL record (Fig. 6b), particularly during the Medieval Climate Anomaly (MCA) when warmer conditions are evident in both datasets. In contrast, a new $\delta^{18}O$ from Norman's Pond in Newfoundland

570 km south of GL, Finkenbinder et al. (2022) shows warmer conditions only during the later stages of the MCA, with anomalously cold conditions around 1000 CE that are less pronounced in the GL record. Despite these differences, there is a notable correlation between the two-time series (Fig. 6c), especially during the last millennium ($r = 0.46$).







**Figure 6: Comparison of the Grand Lake varve thickness (VT) series with regional proxy data. (a) VT at Grand Lake compared to**
**Eastern Canadian tree-ring density. (b) and (c) show the same comparison as (a), but with sea surface temperatures (SST) from**
**north Iceland (b) and δ18O from Norman's Pond in Newfoundland (c). In (a), the records are smoothed using a 21-year running**
**mean for clarity, while in (b) and (c), the records are resampled to match the lowest time resolution (4 years and 10 years,**
**respectively).**





One of the most notable features in the varve thickness series at GL is the major warm peak between the 1150s-1170s CE. This warm anomaly is also evident in the reconstructed summer Northern Hemisphere temperatures based on tree rings (Fig. 7a) and the $\delta^{18}$O record from Norman's Pond (Fig. 6c). Additionally, two reconstructed Atlantic Multidecadal Variability (AMV, Lapointe et al. 2020, Wang et al. 2017) records show a significant correlation with our VT record (Fig. 7b, c), underscoring the strong influence of SST anomalies in the North Atlantic on the GL region. Another prominent feature in the

GL record is the sharp decline in values between 1520-1530s CE, which is also reflected in the SST reconstruction from North Iceland (Fig. 6b), both AMV reconstructions (Fig. 7b,c), and the $\delta^{18}$O record from Norman's Pond (Fig. 6c). However, this decline is not reflected in the summer temperature reconstruction from the MXD network in Eastern Canada, nor in the summer Northern Hemisphere temperature reconstruction, suggesting that SST was the primary driver of this ~1530s decline. Overall, these reconstructions exhibit similar patterns, with a warm Medieval Warm Period (MWP) followed by the cooler conditions

of the Little Ice Age (LIA).





**Figure 7: Comparison between the VT record at GL and Northern Hemisphere high-resolution proxy records (a) spanning the past ~1500 years. (b) and (c) same as (a) but showing two reconstructed AMV (Lapointe et al. 2020, Wang et al. 2017). Time series are filtered by a 21-year running mean.**






The two warm peaks during the late 1300s, as observed in several highly resolved marine records from the Labrador Sea (Lapointe and Bradley, 2021), are also evident in the varve thickness record (Fig. 7b). This shows that this warm episode extended across a broad region, from the western to the eastern portions of the Labrador Sea, a pattern consistent with a persistent negative phase of the NAO. Similar to the reconstructed AMV (Lapointe et al., 2020), varve thickness at Grand

Lake progressively declines following this late 1300s warm event. These parallels suggest that the Labrador region is strongly influenced by ocean-atmosphere interactions across the North Atlantic and that the Grand Lake varve record has captured significant climate variability driven by negative NAO phases over the past ~1500 years.

## 5.4 Teleconnection influencing temperature and precipitation

Investigating possible large-scale teleconnections in the GL region, we find a positive spatial correlation between Goose A

winter temperatures (JFM) and the atmospheric pressure at 500 hPa (z500 hPa) related to an anticyclonic system developed over Greenland and the Canadian Arctic. We also find a negative correlation with lower atmospheric pressure dominating the latitude band 30-40˚N in the Atlantic (Fig. 8a). This pattern is reminiscent of the positive Greenland Blocking index (GBI) (Hanna et al., 2016) (Fig. 8b). We observe that this atmospheric pattern persists across all seasons, though it is less pronounced during the summer. The correlation between winter GBI and Goose A temperature ($r = 0.83$, $p < 0.001$) (Fig. S6) and the NAO

($r = -0.77$, $p < 0.001$) confirms that the site is influenced by the GBI.




**Figure 8: (a) Spatial correlation between Goose A and 500hPa geopotential height anomaly (Dee et al. 2011). (b) same as above but for the Greenland Blocking and 500hPa.**


Periods with high winter precipitation at Goose A coincides with high atmospheric pressure over Greenland and south of Iceland (Fig. 9a). The composite map of daily precipitation > 10 mm also captures this atmospheric anomaly and an overall high-pressure over much of the Canadian Arctic and west Greenland; a pattern reminiscent of the negative phase of the NAO (Fig. 9b). This is supported by a significant correlation between GBI (or NAO) and Goose A precipitation in winter (Fig. 9c)

(Fig. S7).



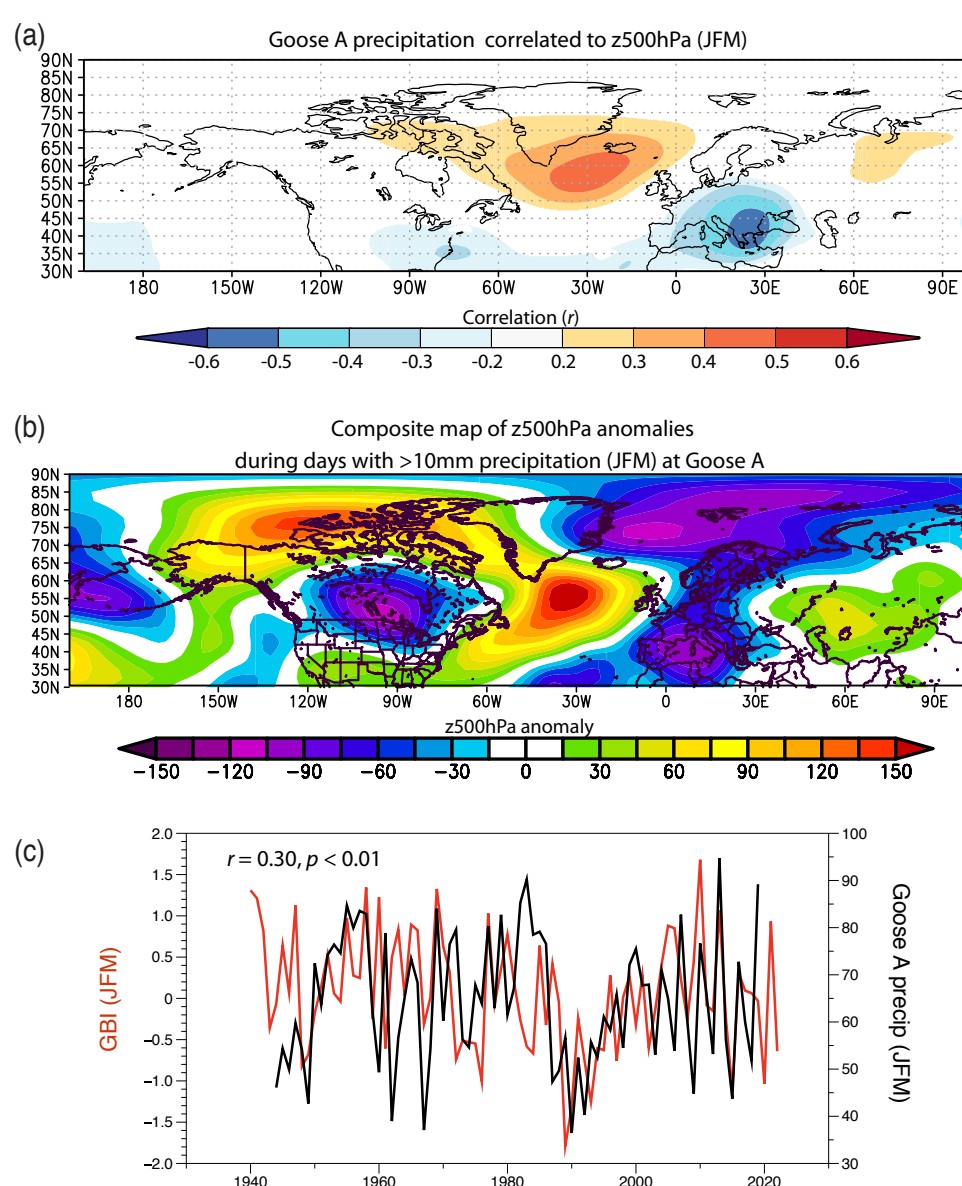

**Figure 9: (a) Spatial correlation between winter (JFM) precipitation at Goose A from the Global Historical Climatology Network and 500 hPa geopotential height from Era-Interim (Dee et al. 2011). (b) Composite map of extreme winter precipitation (>10 mm d⁻¹) at Goose A. (c) Goose A winter precipitation compared to Greenland Blocking index.**

The blocking pattern over the North Atlantic observed with spatial correlation between Goose A winter precipitation and atmospheric pressure (500 hPa) (Fig. 9a,b) supports the idea that snowfall in the Grand Lake area is, to some extent, influenced



by GBI (and NAO). The positive phase of the GBI (negative phase of the NAO) is associated with warmer temperatures in
Labrador (Shabbar et al., 1997) and higher than normal winter precipitation (Bonsal and Shabbar, 2008). During GBI+ and
NAO-, a high-pressure anticyclonic system prevails over the southeast tip of Greenland and is associated with weaker
westerlies and an increase in meandering of the jet stream (Shabbar et al., 2001). These blocking patterns allow a north-easterly
shift in the eastern Canada storm track toward the study region and higher occurrence of cyclones (storms) and more
precipitation over south and central Labrador (Chartrand and Pausata, 2020). This change in the preferred path of storms exerts
an influence on Labrador climate particularly during the cold season, leading to an increase in the snow accumulation and
consequently to higher river discharge in the region. Since Grand Lake varves are sensitive to snow, i.e., warm and snowy
winters are reflected by thicker varves, it is logical to observe a relation between GBI (NAO) and precipitation reconstructed
by our annually resolved sedimentary record.

While GBI annually resolved reconstructions beyond the instrumental record have not been generated yet, many NAO
reconstructions exist. However, when these reconstructed NAO are compared individually, they generally do not agree well
before the instrumental era (Pinto and Raible, 2012). It has been established that the frequency of Greenland atmospheric
blocking tends to be more persistent during the positive phase of the AMV (Hahn et al., 2018; Hakkinen et al., 2011; Peings
and Magnusdottir, 2014). The AMV exerts an influence on Labrador by adjusting the transfer of heat from the warm ocean to
cold overlying air throughout the cold season, leading to increase snow precipitation in the area. The overall similarities
between the reconstructed AMVs and the proxy records in this study point to an atmospheric teleconnection, which we suggest
is driven by NAO variations modulated by the underlying AMV state.

**5.5 Recent history of the NAO in the context of the last 1500 years**

As Arctic sea ice is projected to decrease in the coming decades, three circulation patterns are expected to become more
frequent under low sea ice conditions: Scandinavian, Ural, and Greenland Blocking patterns (Craseman et al. 2017, Tedesco
et al. 2020, Lapointe et al. 2024). Since 2005, the intensification of summer Greenland Blocking has played a significant role
in accelerating surface melting over the Greenland Ice Sheet and reducing Arctic sea ice extent. However, it is important to
consider seasonal differences in Greenland Blocking and NAO behavior. During winter, the positive phase of the NAO has
shown a marked increase since the late 1980s and early 1990s (Fig. S8), leading to stronger westerly winds. This shift coincides
with a reduction in sediment influx at Grand Lake, which is also reflected in precipitation data from nearby weather stations
(Fig. 5). A decreasing trend can be observed in the Grand Lake varve thickness record over the past 50 years, but it is not
unusual in the context of the past 1500 years. Importantly, the trend in winter for the NAO time series is toward more positive
values (Fig. S8), and climate models generally predict an increase in the winter NAO index under future high-emissions
scenarios (McKenna and Maycock 2022, Lee et al. 2021, Gillet and Fyfe 2013). In addition, with the anticipated increase in
melt from the Greenland Ice Sheet and other high-latitude glaciers, the influx of freshwater into the North Atlantic is expected
to maintain a positive NAO in winter (Oltmanns et al., 2020), which in turn would lead to a reduction of the snowpack in the
area. According to the information from our long-term record, the positive trends (NAO+), if continued, will likely act to

reduce winter precipitation in the region of Grand Lake, with potential implications for hydroelectricity. Future research should focus on these modes of variability and the mechanisms driving their evolution, highlighting the need for more proxy records that capture these dynamics.

**6 Conclusions**

In this paper we present a new hydroclimatic record at annual resolution in eastern North America and the first millennial reconstruction based on varved sediments from a deep fjord lake at the western fringe of the Atlantic Ocean. The annual character of the 1523 year-long lamination sequence from the distal core GL-13 has been confirmed. The Medieval Climate Anomaly (~1050–1225 CE) is characterized by thicker varves indicative of higher total precipitation, while the Little Ice Age

(1400-1875 CE) recorded thinner varves indicative of a dryer climate. Present-day teleconnection highlights that the precipitation in the region is in part modulated by the GBI+ (or negative NAO), which is mediated by oceanic-atmospheric processes associated with the AMV. The similarities between our record and others suggest that the teleconnection, specifically the negative phase of the North Atlantic Oscillation (NAO), has been a persistent feature of regional hydroclimate over the past 1000+ years. This positions the GL record as a promising candidate for future NAO/AMV reconstructions. Importantly,

the decreasing trend in the Grand Lake varve thickness over the past 50 years is not unusual when viewed within the context of the last 1500 years. However, it highlights that if the winter NAO continues its positive trend in the future, it could have major implications for hydroelectricity production from this region.

**Data availability**

Varve thickness dataset will be uploaded in the Varved Sediments Database (VARDA), https://varve.gfz-potsdam.de/

**Author contributions**

**François Lapointe**: Conceptualization; Formal analysis; Investigation; Validation; Visualization; Writing - original draft; Writing - review & editing. **Antoine Gagnon-Poiré**: Conceptualization; Methodology; Formal analysis; Investigation; Validation; Data curation; Writing - original draft; Writing - review & editing; Visualization. **Pierre Francus**: Conceptualization; Data curation; Methodology; Ressources; Writing - review & editing; Visualization; Supervision; Project

administration; Funding acquisition; **Patrick Lajeunesse**: Conceptualization; Ressources; Writing - review & editing. **Clarence Gagnon**: Formal analysis; Investigation.

**Declaration of competing interest**

The author Pierre Francus is member of the editorial board of Climate of the Past



**Acknowledgements**

The authors are grateful to Arnaud De Coninck, David Deligny, David Fortin and Louis-Frédéric Daigle for their participation during fieldwork and in the laboratory. We greatly thank Wanda and Dave Blake from North West River for their guiding experience and accommodation at Grand Lake. We thank the Labrador Institute at North West River for the use of their facility during fieldwork, and R.S. Bradley for lending the vibracoring equipment. We want to thank Stéphane Ferré from the Micro-Geoarchaeology Laboratory of the Centre d'Études Nordiques (CEN) in Québec, QC, Canada, for the production of high-

quality thin sections used in this study. Finally, many thanks to Monique Gagnon and Charles Smith for reviewing the English of an earlier version of the paper.

**Financial support**

This work was supported by an NSERC Discovery grants (RGPIN-2014-05810 and RGPIN-2019-06593 awarded to PF, and by a US National Science Foundation grant from the P4Climate program (NSF # 2402628) awarded to FL.




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
