# Peer review of "A new 1500-year-long varve thickness record from Labrador, Canada, uncovers significant insights into large-scale climate variability in the Atlantic"

_EGUsphere, 2025_

## Author Comment (AC1)

**Reply to RC1: reviewer comments in blue, and our reply in black.**

This paper presents a valuable new hydroclimatic record derived from varved sediments in a deep fjord lake at the western edge of the Atlantic Ocean, offering insights into climate variability over the past 1500 years. The authors emphasize the potential of the GL-13 lamination sequence as a proxy for regional hydroclimate and suggest that their findings align with broader teleconnections, particularly with the North Atlantic Oscillation (NAO) and the Atlantic Multidecadal Variability (AMV). While the paper makes important contributions to our understanding of long-term hydroclimatic dynamics, there are several notable gaps in the analysis that warrant further attention. The interpretation of this high-potential record is undervalued.

Thank you for your good words about the high potential of this site and our study.

**1. Lack of Seasonality Analysis:**

One missing issue is the absence of a detailed evaluation of seasonality within the varve record. The authors assert that the varves are annual in nature, with thicker varves indicating higher precipitation and thinner varves associated with drier periods. However, the seasonal distribution of precipitation within the year (e.g., whether the wetter periods are associated with specific seasons such as winter or summer) is not discussed. Given the potential sensitivity of the region's climate to changes in seasonal patterns, a deeper understanding of seasonality could offer important insights into how different climatic factors may influence precipitation timing and intensity throughout the year. It could have important implications for the interpretation of historical hydroclimatic changes and the impacts of climate variability on local ecosystems.

There seasonality of the varves proximal to the main tributaries is more pronounced and was detailed in the paper by Gagnon-Poiré et al. (2021): varve thickness is mainly driven by the spring discharge. However, as supplementary figures S4 and S5 show, the spring discharge is driven by snow melt, the latter being controlled by winter precipitations. On line 184, we clearly indicate that snow precipitation is the main driver of the varve thickness record, that means that varve record a winter signal. Yet, all the teleconnections subsequently discussed (Figs 8 and 9) are focusing on the winter month (JFM). Therefore we are not sure what we need to do to get a "deeper understanding of seasonality."

**2. Teleconnection Mechanisms:**

I appreciate the attempt to come up with spacial patterns. While the authors suggest that Greenland blocking and NAO plays a central role in modulating precipitation in the study region, they do not provide a comprehensive evaluation of how this teleconnection operates within the context of proxy record. Also the associated time and spacial scale is not considered.

The precipitation data from Goose A shows a sensitivity to North Atlantic Oscillation (NAO) changes, impacting both temperature and precipitation variability. The significant correlation between the varve thickness series and precipitation suggests that this proxy record is, to some extent, responsive to NAO fluctuations. As illustrated in supplementary figures S4 and S5 and as mentioned above, spring discharge is primarily influenced by snowmelt, which is directly connected to winter precipitation. Therefore, we can associate the atmospheric patterns linked to increased precipitation with more pronounced negative NAO phases.

Furthermore, understanding whether the relationship between the record and the pattern is constant across different phases would be valuable for refining the interpretation of the varve record.

Evaluating the persistence of this teleconnection over time and across various scales is challenging, as the different reconstructed NAO indices lack correlation before the instrumental period. This discrepancy is likely influenced by seasonality; tree-ring records are more closely associated with summer temperatures, suggesting that earlier reconstructed NAOs using tree rings may be more relevant to summer conditions. In contrast, the NAO is primarily an atmospheric pattern linked to winter variability, making our proxy record a unique resource for understanding past winter NAO dynamics.

We believe our record will be invaluable for future NAO reconstructions within a spatial network of sensitive winter NAO proxies. The scope of this study is to present the first 1,500 years of hydroclimate variability in northeastern Canada, detailing the age model, local comparisons, and potential teleconnections. An additional manuscript is in progress that will further explore the record across time and scale, as well as the underlying mechanisms.

**3. Implications for Hydroelectricity and Future Climate Trends:**

The paper makes a useful connection between past climate variability and potential drivers. Nnoting that the trend of decreasing varve thickness over the past 50 years is consistent with long-term variability, it would be interesting to make the link to climate model output.

Yes, making the link with climate models input would be very useful indeed, and we have the plan to look into this in a subsequent paper, but this is beyond the goal of this paper.

**4. Further Data Validation and Comparative Analysis:**

The paper claims that the GL-13 varve sequence is robust and confirms its annual character over 1523 years. The authors briefly mention similarities between their record and others, but a more in-depth comparative analysis would strengthen the argument that the GL-13 sequence is a reliable proxy for regional hydroclimate and enhances the credibility of their interpretation.

There are not that many records out there that are geographically close enough to be under the same hydroclimatic influence and they don't have the same resolution as ours.

The eastern Canadian tree-ring record (Figs 1 and 6) is a summer temperature record, so not directly comparable to ours. Moreover, the hydrological regimes in western Quebec are different from those close to the Labrador coast (Nasri et al. 2020).

The Norman's Pound record in Newfoundland is a complex stable isotope record that is influenced by precipitations and temperature, both with a seasonal effect on the isotopic values. We outline that the LIA and the MCA were recorded in Norman's Pound as it is the case at Grand Lake, but it seems it is the only link that we can do.

Nevertheless, we agree that such suggested comparison would strengthen the credibility of our interpretation but we are not aware of any record that would be appropriate for that, except the ones already used in the paper. Any suggestion is, of course, welcome.

**Conclusion:**

While this paper presents a valuable new hydroclimatic record and offers interesting insights into long-term hydroclimate variability in eastern North America, it would benefit from a more nuanced discussion of the seasonality of precipitation and a deeper evaluation of the teleconnection mechanisms driving the

observed changes in the varve thickness. By addressing these gaps, the authors could provide a more comprehensive understanding of the region's hydroclimatic dynamics and enhance the broader implications of their work, particularly with regard to future climate change and its potential impact.

We will modify the final revised paper to add more details about the mechanisms between varve thickness, the seasonality of precipitations and their teleconnections with the major modes of climate variability.

**References**

Nasri B.R., Boucher E., Perreault L., Remillard B.N., Huard D., Nicault A. & Projects A.-P. (2020). Modeling Hydrological Inflow Persistence Using Paleoclimate Reconstructions on the Quebec-Labrador (Canada) Peninsula. *Water Resources Research* 56, 5: e2019WR025122.

---

## Author Comment (AC2)

The authors present a well-written and insightful paper that proposes to use a clastic varve records covering over 1,500 yrs to track past hydroclimates in northern America. By correlating the record of varve thickness to regional precipitation patterns as well as other regional records, the variations observed are attributed to changes in winter snowfall. These interpretations are then placed in a wider context to determine the role of large-scale synotic climate, such as the North-Atlantic Oscillation and Greendand Blockage on winter precipitation. The 1,500 yr long varved record is therefore suggested to provide insights into past NAO/GB variability.

While this is undoubtly a valuable record and a publication that falls within the scope of Climate of the Past, I would like to raise a few issues regarding the predicting power of the record to track winter precipitation and NAO. I do not refute the conclusions *per se*, but I question the robustness of the statistical methods on which the argumentation relies and try to propose additionnal ways to explore the data.

Thank you very much for this general comment.

**Varves**:
- I know this is standard in varve research, but there is no mention that the sediment blocks were embedded in resin. Since this paper is adressing a larger audience, I would recommend to add a line on that. Also are the SEM pictures taken on the thin sections or sediment blocks?

  Yes, the thin sections were made using epoxy resin. Thanks for your point, we will add clarification about it.

- I don't doubt that these laminations are real varves (indeed a textbook example of clastic varves!) but I was wondering whether you had any hint of years with more than two sublayers, i.e., with more than one discharge event?

Yes, additional layers are very clear in the cores proximal to the two main tributaries. This was reported in our previous paper (Gagnon-Poiré *et al.* 2021). In this distal site, there are a few (3-4) intercalated coarser layers, but their origin cannot be attributed, for sure, to additional discharge events.

**Data analysis**:
- Determination of drivers of sediment input (VT). Correlation of precipitation at Goose A and sediment accumulation (VT) (fig 5): Please provide $r^2$ instead of (or in addition to) r. The r value shows that the variables are positively and significantly correlated (aka. yes, total precipitation/snowfall influences the amount of sediment deposited at the site). The $r^2$ value shows how much of the variance in the dataset can be explained by the driver considered – since you are using this relationship to make predictions about winter precipitation and the NAO, it is not trivial. In your case, an $r^2$ of ca. 0.1 (which is more or less what you must have here) means that most of the variance in the dataset (90%) is explained by something else than total precipitation or snow precipitation. Did

you investigate other drivers? I am wondering whether running a multiple linear regression with adjusted $r^2$ would help to constrain the role of e.g., rain+snow+temperature on sediment input. In terms of processes, I can well imagine that a combination between how fast the temperature rises in the spring and how much snow has been deposited in the winter might exert a strong control on discharge and flow strength.

That's an excellent point. We will include $R^2$ values in the figures alongside the correlation coefficients (R values). Our multiple linear regression analysis indicates stronger correlations and more significant p-values for the period 1972–2017. The reason for this is unclear, but it may be related to a change in meteorological instrumentation following the 1970s. We attempted to obtain more information about the meteorological station at Goose Bay but have not yet been successful. In any case, the strongest correlation observed is with snow deposition.

From 2017-1943:

```
Call:
lm(formula = VT ~ SNOW_Nov_May + temp + raingoosebay, data = df_scaled_subset)

Residuals:
     Min      1Q   Median       3Q      Max
-2.15641 -0.66563 -0.03165  0.42732  2.65285

Coefficients:
               Estimate Std. Error t value Pr(>|t|)
(Intercept)  -3.057e-17  1.093e-01   0.000  1.00000
SNOW_Nov_May  3.258e-01  1.129e-01   2.887  0.00515 **
temp          1.636e-01  1.157e-01   1.414  0.16164
raingoosebay  8.733e-02  1.154e-01   0.757  0.45171
* * *
Signif. codes:  0 '***' 0.001 '**' 0.01 '*' 0.05 '.' 0.1 ' ' 1

Residual standard error: 0.947 on 71 degrees of freedom
Multiple R-squared:  0.1396,   Adjusted R-squared:  0.1032
F-statistic: 3.839 on 3 and 71 DF,  p-value: 0.01318
```

From 2017-1972:

```
Call:
lm(formula = VT ~ SNOW_Nov_May + temp + raingoosebay, data = df_scaled_subset)

Residuals:
    Min      1Q  Median      3Q     Max
-1.4210 -0.7043 -0.1026  0.4959  2.2126

Coefficients:
              Estimate Std. Error t value Pr(>|t|)
(Intercept)  4.393e-17  1.308e-01   0.000  1.00000
SNOW_Nov_May 4.017e-01  1.337e-01   3.004  0.00443 **
temp         1.199e-01  1.439e-01   0.833  0.40931
raingoosebay 2.288e-01  1.433e-01   1.596  0.11780
* * *
Signif. codes:  0 '***' 0.001 '**' 0.01 '*' 0.05 '.' 0.1 ' ' 1

Residual standard error: 0.8965 on 43 degrees of freedom
Multiple R-squared:  0.2486,   Adjusted R-squared:  0.1962
F-statistic: 4.743 on 3 and 43 DF,  p-value: 0.006044
```

- Same comment for Figure 9c: use r2 instead of r if you want to show that GB is a good predictor for winter precipitation. By the way: how is the GBI calculated?

OK, we'll do. We will add a sentence that describes the GBI along with the reference from Hanna et al. 2016. Thanks.

- Minor point: in Fig 5, did I understand correctly that you compare log(VT) to log(precip)? That would make sense since both datasets are constrained (R+, only positive).

The data are only normalized (relative to the mean and the standard deviation).

- p values need to be reported and discussed (see for instance the recent paper of *Benjamin et al., Nature Human Behaviour 2018*)

OK, we will discuss p-values further.

- time-series analyses: the data have been analysed using a wavelet analysis (Fig. S2) (using which software?). I am a bit surprised that the analysis (which is unique as being annually-resolved) shows little high-frequency variability, esp. in the range of those originating from the NAO (7-yr if I am right?), if this is supposed to be an important driver. Do the authors have an explanation for this? Did they also perform a periodogram (e.g., multi-tapper)? Did you run time-series analyses on the xrf records?

There is actually a period of significant variability centered around 7 years (>99% conf. level: see periodogram below), which is also the case when the data are log transformed (lower panel). This is also observed in a new wavelet analysis that is better detailed (below).
No, we haven't run time series analyses on the XRF data, which is not the scope of this paper.

[Figure]

[Figure]

Fig. Upper panel: Periodogram of the varve thickness (VT) record from Grand Lake. The periodogram displays the distribution of spectral power across different frequencies, revealing dominant periodic components in the time series. To assess the statistical significance of spectral peaks, the results are compared against a null hypothesis that the time series follows a red-noise (AR(1)) process, which represents a baseline of natural variability with no true periodicity. Peaks that exceed the 95% confidence threshold—determined from the theoretical red-noise spectrum— are considered statistically significant, indicating the presence of persistent periodic signals unlikely to result from

random fluctuations alone. Lower panel : same as the upper panel, but analysis of the log varve thickness. Many spectral peaks exceed the 95% confidence level (including at a 7-year cycle), suggesting that they reflect periodic signals.

**Wavelet Power Spectrum of Varve Thickness**

[Figure]

**Wavelet Power Spectrum of log VT**

[Figure]

Fig. Wavelet analysis of the 1,500-varve record from Grand Lake, Labrador. White contours delimit areas of >95% confidence levels. Statistical significance was assessed using 100 Monte Carlo simulations were performed to estimate the distribution of wavelet power under a red-noise (AR(1)) null model. The black dot represents a grid cell where the observed power exceeds the significance threshold; they are helpful for visually spotting localized significant features within the broader white contour region. Short periods of around 7-year cycles were particularly active during 600-800AD, the Medieval climate anomaly (1100-1380) and the latter part of the Little Ice Age.

In terms of structure, the authors might consider moving Fig. 5 and the associated paragraphs to the discussion, after section 5.1. The attemps to determine the drivers will come more logically after having explained the depositional process of the varves.

This is a great comment. We will move this part to the discussion. Thanks

**Interpretations:**
- Line 235: "One of the most notable features in the varve thickness series at GL is the major warm peak between the 1150s-1170s CE." I don't understand this sentence. In Fig. 5, the only predictor tested is precipitation, not temperature. The correlations with temperature proxies (not temperature records) only tell us that general trends in hydroclimates might be consistent regionally. Please rephrase accordingly.

That is correct; however, paleo evidence suggests this period was likely characterized by elevated temperatures and higher sea surface temperatures (SSTs). This is supported by comparisons between our record and reconstructed Northern Hemisphere temperature datasets, including those reflecting AMV phases. These findings suggest that increased precipitation during this time was likely driven by warmer regional and/or hemispheric temperatures. Nevertheless, we will rephrase the sentence accordingly.

- As in most fluvial systems, the VT record might be affected by memory/autocorrelation effects (and nonlinear relationships between precipitation/flood strength/sediment transport). Did you try to use differentiation ($y_{diff}=y-y_{-1}$) to reduce these effects? Or investigate the autocorrelation patterns?

The VT time series shows strong autocorrelation at lags extending up to 40 years (left panel). When applying the same analysis to the differenced series (right panel: $Y_t = X_t - X_{t-1}$), the data appear more random, with values falling within the dashed confidence bounds. This indicates that the original autocorrelation likely stems from a long-term trend rather than true cyclicity. There is strong evidence from paleoclimate records suggesting this trend may be linked to long-term oceanic variability, such as the Atlantic Multidecadal Variability (AMV). Additionally, spectral analysis reveals significant multidecadal variability, with notable peaks around 48 and 76 years (which could be linked to AMV).

[Figure]

Fig. Autocorrelation functions (ACF) of the original and differenced varve thickness (VT) record from Grand Lake over the past 1,500 years. The ACF of the original VT series (left panel) shows positive autocorrelation at short lags, with values remaining above the 95% confidence bounds up to 40 years. This indicates the presence of low-frequency persistence, likely related to long-term climate variability. In contrast, the ACF of the first-differenced VT series (right panel) displays values that mostly fall within the 95% confidence bounds, consistent with white noise. This suggests that the autocorrelation observed in the original series is primarily driven by long-term trends rather than cyclic behaviour.

- I found the discussion on weather patterns interesting and have no comments on that part.

Thank you.

- A purely curiosity-driven question: do the authors notice changes in interannual variability of VT (potentially precipitation) throughout the record?

Detecting changes in year-to-year variability is challenging due to the high level of noise in the varve thickness record. We think that the NAO plays a significant role, as evidenced by the marked reduction in precipitation during the 1990s, which coincided with a pronounced intensification of the NAO+. We are currently preparing a second paper that applies change point analysis to identify and characterize abrupt shifts in the time series. This work also aims to examine how these shifts align in timing and phase with other high-resolution proxy records from the subpolar gyre.

**Figures:**
Figure 2: pannel b is hard to see in an otherwise well-constructed and clear figure. Pannel c: what do the codes mean (abp, abo, ab…?)

We will better explain it in the figure caption. Thanks

In Fig. 5,6,7: please add underlines or boxes to show where the MCA and the LIA are.
Figure 6: please plot the original data with the resampled ones so that one can keep track of the data transformation.

Ok, we'll do.

**Technical comment**: There is ample litterature to emphasize the need to use a log-ratio transformation of xrf data (which are by essence compositional and therefore constrained, see for instance *Kucera and Malmgren, Marine Micropaleontology 1998*). Data presented as count rates can provide spurious results and should therefore be transformed (see recent publication by *Bertrand et al., Earth Science Reviews 2024*, with many clear examples).
I would therefore recommend that the authors revise figure 2 to show the iron relative concentration as log ratios (or centered log-ratios), with a clear x-axis and unit given.

We are perfectly aware that central log-ratios (clr) are the best way to use the XRF compositional datasets. However, clr are needed to help interpret variations of a group of elements over an entire sediment sequence and to deal with potential biases due to, for instance, compaction. Here, we only use Fe peaks to help identify the clay caps of the varves over a very short interval. In this very case, we don't think that log-ratios will provide spurious results. We are therefore respectfully declining the invitation to revise figure 2.

**Minor mistakes:**
Line 156: Three years *were* added
Thanks

---

## Author Response (AR2)

The reviewer's comments are in black and our replies are in green. Line numbers refer to the annotated manuscript.

**REVIEWER 1 :**

**1. Lack of Seasonality Analysis:**

One missing issue is the absence of a detailed evaluation of seasonality within the varve record. The authors assert that the varves are annual in nature, with thicker varves indicating higher precipitation and thinner varves associated with drier periods. However, the seasonal distribution of precipitation within the year (e.g., whether the wetter periods are associated with specific seasons such as winter or summer) is not discussed. Given the potential sensitivity of the region's climate to changes in seasonal patterns, a deeper understanding of seasonality could offer important insights into how different climatic factors may influence precipitation timing and intensity throughout the year. It could have important implications for the interpretation of historical hydroclimatic changes and the impacts of climate variability on local ecosystems.

> The seasonality of the varves proximal to the main tributaries is more pronounced and was detailed in the paper by Gagnon-Poiré et al. (2021): varve thickness is mainly driven by the spring discharge. However, as supplementary figures S3 and S4 show, the spring discharge is driven by snow melt, the latter being controlled by winter precipitations. On line 268, we clearly indicate that snow precipitation is the main driver of the varve thickness record, which means that varves record a winter signal. Yet, all the teleconnections subsequently discussed (Figs 9 and 10) are focusing on the winter month (JFM). Seasonality and teleconnections are also more extensively discussed in lines 469-489.

**2. Teleconnection Mechanisms:**

I appreciate the attempt to come up with spacial patterns. While the authors suggest that Greenland blocking and NAO play a central role in modulating precipitation in the study region, they do not provide a comprehensive evaluation of how this teleconnection operates within the context of proxy record. Also the associated time and spacial scale is not considered.

> The precipitation data from Goose A shows a sensitivity to North Atlantic Oscillation (NAO) changes, impacting both temperature and precipitation variability. The significant correlation between the varve thickness series and precipitation suggests that this proxy record is, to some extent, responsive to NAO fluctuations. As illustrated in supplementary figures S3 and S4 and as mentioned above, spring discharge is primarily influenced by snowmelt, which is directly connected to winter precipitation. Therefore, we can associate the atmospheric patterns linked to increased precipitation with more pronounced negative NAO phases.

Furthermore, understanding whether the relationship between the record and the pattern is constant across different phases would be valuable for refining the interpretation of the varve record.

> Evaluating the persistence of this teleconnection over time and across various scales is challenging, as the different reconstructed NAO indices lack correlation before the

instrumental period. This discrepancy is likely influenced by seasonality; tree-ring records are more closely associated with summer temperatures, suggesting that earlier reconstructed NAOs using tree rings may be more relevant to summer conditions. In contrast, the NAO is primarily an atmospheric pattern linked to winter variability, making our proxy record a unique resource for understanding past winter NAO dynamics. This is now clearly stated in our manuscript with new lines 469-489.

We believe our record will be invaluable for future NAO reconstructions within a spatial network of sensitive winter NAO proxies. The scope of this study is to present the first 1,500 years of hydroclimate variability in northeastern Canada, detailing the age model, local comparisons, and potential teleconnections. An additional manuscript is in progress that will further explore the record across time and scale, as well as the underlying mechanisms.

**3. Implications for Hydroelectricity and Future Climate Trends:**

The paper makes a useful connection between past climate variability and potential drivers. Noting that the trend of decreasing varve thickness over the past 50 years is consistent with long-term variability, it would be interesting to make the link to climate model output.

Yes, making the link with climate model input would be very useful indeed, and we have the plan to look into this in a subsequent paper, but this is beyond the goal of this paper.

**4. Further Data Validation and Comparative Analysis:**

The paper claims that the GL-13 varve sequence is robust and confirms its annual character over 1523 years. The authors briefly mention similarities between their record and others, but a more in-depth comparative analysis would strengthen the argument that the GL-13 sequence is a reliable proxy for regional hydroclimate and enhances the credibility of their interpretation.

There are not that many records out there that are geographically close enough to be under the same hydroclimatic influence and they don't have the same resolution as ours. The eastern Canadian tree-ring record (Figs 1 and 7a) is a summer temperature record, so not directly comparable to ours. Moreover, the hydrological regimes in western Quebec are different from those close to the Labrador coast (Nasri et al. 2020).

The Norman's Pound record in Newfoundland is a complex stable isotope record that is influenced by precipitations and temperature, both with a seasonal effect on isotopic values. We outline that the LIA and the MCA were recorded in Norman's Pound, as it is the case at Grand Lake, but it seems it is the only link that we can do.

Nevertheless, we agree that such suggested comparison would strengthen the credibility of our interpretation, but we are not aware of any records that would be appropriate for that, except the ones already used in the paper. Any suggestion is, of course, welcome.

**Conclusion:**

While this paper presents a valuable new hydroclimatic record and offers interesting insights into long- term hydroclimate variability in eastern North America, it would benefit from a more nuanced discussion of the seasonality of precipitation and a deeper evaluation of the teleconnection mechanisms driving the observed changes in the varve thickness. By addressing these gaps, the authors could provide a more comprehensive understanding of the region's hydroclimatic dynamics

and enhance the broader implications of their work, particularly with regard to future climate change and its potential impact.

> We improved the discussion about this topic in section 5.5, better explaining the seasonality of precipitations and their teleconnections with the major modes of climate variability, and underlining that our record is driven by winter conditions.

**REVIEWER 2 :**

**Varves**:

I know this is standard in varve research, but there is no mention that the sediment blocks were embedded in resin. Since this paper is adressing a larger audience, I would recommend to add a line on that. Also are the SEM pictures taken on the thin sections or sediment blocks?

> Lines 100-101: Normandeau et al. (2019) explains the whole protocol, including subsampling and impregnation. We nevertheless added some details of the different steps.
> After verification, the images were taken from the thin sections, as indicated on line 97. We have not changed anything here.

I don't doubt that these laminations are real varves (indeed a textbook example of classic varves!) but I was wondering whether you had any hint of years with more than two sublayers, i.e., with more than one discharge event?

> Yes, additional layers are very clear in the cores proximal to the two main tributaries. This was reported in our previous paper (Gagnon-Poiré *et al*. 2021). In this distal site, there are a few (3- 4) intercalated coarser layers that were used as marker beds to correlate the two cores of this distal site, but their origin cannot be attributed, for sure, to additional discharge events. We don't think it is necessary to add something about this in the paper.

**Data analysis**:

Determination of drivers of sediment input (VT). Correlation of precipitation at Goose A and sediment accumulation (VT) (fig 5): Please provide $r^2$ instead of (or in addition to) r. The r value shows that the variables are positively and significantly correlated (aka. yes, total precipitation/snowfall influences the amount of sediment deposited at the site). The $r^2$ value shows how much of the variance in the dataset can be explained by the driver considered – since you are using this relationship to make predictions about winter precipitation and the NAO, it is not trivial. In your case, an $r^2$ of ca. 0.1 (which is more or less what you must have here) means that most of the variance in the dataset (90%) is explained by something else than total precipitation or snow precipitation. Did you investigate other drivers? I am wondering whether running a multiple linear regression with adjusted $r^2$ would help to constrain the role of e.g., rain+snow+temperature on sediment input. In terms of processes, I can well imagine that a combination between how fast the temperature rises in the spring and how much snow has been deposited in the winter might exert a strong control on discharge and flow strength.

> That's an excellent point. We included $r^2$ values in figures 6, 7 and 9 alongside the correlation coefficients (r values).
>
> We tested the multiple linear regression analyses. It indicates stronger correlations and more significant p-values for the period 1972–2017 (see below). The reason for this is unclear, but it may be related to a change in meteorological instrumentation following the

1970s. We attempted to obtain more information about the meteorological station at Goose Bay but have not yet been successful. In any case, the strongest correlation observed is with snow deposition. Since this analysis raises more questions than solving any, or improves substantially the correlations, we have not changed the manuscript about this topic.

From 2017-1943:

```
Call:
lm(formula = VT ~ SNOW_Nov_May + temp + raingoosebay, data = df_scaled_subset)

Residuals:
     Min      1Q   Median       3Q      Max
-2.15641 -0.66563 -0.03165  0.42732  2.65285

Coefficients:
              Estimate Std. Error t value Pr(>|t|)
(Intercept) -3.057e-17  1.093e-01   0.000  1.00000
SNOW_Nov_May 3.258e-01  1.129e-01   2.887  0.00515 **
temp         1.636e-01  1.157e-01   1.414  0.16164
raingoosebay 8.733e-02  1.154e-01   0.757  0.45171
* * *
Signif. codes:  0 '***' 0.001 '**' 0.01 '*' 0.05 '.' 0.1 ' ' 1

Residual standard error: 0.947 on 71 degrees of freedom
Multiple R-squared:  0.1396,   Adjusted R-squared:  0.1032
F-statistic: 3.839 on 3 and 71 DF,  p-value: 0.01318
```

From 2017-1972:

```
Call:
lm(formula = VT ~ SNOW_Nov_May + temp + raingoosebay, data = df_scaled_subset)

Residuals:
    Min      1Q  Median      3Q     Max
-1.4210 -0.7043 -0.1026  0.4959  2.2126

Coefficients:
              Estimate Std. Error t value Pr(>|t|)
(Intercept)  4.393e-17  1.308e-01   0.000  1.00000
SNOW_Nov_May 4.017e-01  1.337e-01   3.004  0.00443 **
temp         1.199e-01  1.439e-01   0.833  0.40931
raingoosebay 2.288e-01  1.433e-01   1.596  0.11780
* * *
Signif. codes:  0 '***' 0.001 '**' 0.01 '*' 0.05 '.' 0.1 ' ' 1

Residual standard error: 0.8965 on 43 degrees of freedom
Multiple R-squared:  0.2486,   Adjusted R-squared:  0.1962
F-statistic: 4.743 on 3 and 43 DF,  p-value: 0.006044
```

Same comment for Figure 9c: use r2 instead of r if you want to show that GB is a good predictor for winter precipita6on. By the way: how is the GBI calculated?

$R^2$ has been added to figure 9c. On line 425, we added a sentence that describes how the GBI is calculated with the reference from Hanna et al. 2016.

Minor point: in Fig 5, did I understand correctly that you compare log(VT) to log(precip). That would make sense since both datasets are constrained (R+, only positive).

The data are only normalized (relative to the mean and the standard deviation). We added that information in the figure caption (now Figure 6).

p values need to be reported and discussed (see for instance the recent paper of Benjamin et al., Nature Human Behaviour 2018)

> p-values were reported in all our correlations.

Time-series analyses: the data have been analysed using a wavelet analysis (Fig. S2) (using which software?). I am a bit surprised that the analysis (which is unique as being annually-resolved) shows little high-frequency variability, esp. in the range of those originating from the NAO (7-yr if I am right?), if this is supposed to be an important driver. Do the authors have an explanation for this? Did they also perform a periodogram (e.g., multi-tapper)? Did you run time-series analyses on the xrf records?

> There is actually a period of significant variability centred around 7 years (>99% conf. level: see periodogram below), which is also the case when the data are log transformed (lower panel). This is also observed in a new wavelet analysis that is better detailed (below).
> We have now included these analyses in the body of the manuscript in section 4.4. Spectral content.
> We specified the R package used in this method section and a periodogram performed on the VT dataset in the main text, better highlighting the significant periodicities (now Fig. 5). The output of these analyses is presented in new lines 229-249. Below are also presented the analyses made on the logVT datasets, but those were not inserted in the revised manuscript because they were showing similar results.
> We haven't run time series analyses on the XRF data, which is not the scope of this paper.

[Figure]

[Figure]

Fig. 5. Upper panel: Periodogram of the varve thickness (VT) record from Grand Lake. The periodogram displays the distribution of spectral power across different frequencies, revealing dominant periodic components in the time series. To assess the statistical significance of spectral peaks, the results are compared against a null hypothesis that the time series follows a red-noise (AR(1)) process, which represents a baseline of natural variability with no true periodicity. Peaks that exceed the 95% confidence threshold—determined from the theoretical red-noise spectrum—are considered statistically significant, indicating the presence of persistent periodic signals unlikely to result from random fluctuations alone. Lower panel : same as the upper panel, but analysis of the log varve

thickness. Many spectral peaks exceed the 95% confidence level (including at a 7-year cycle), suggesting that they reflect periodic signals.

**Wavelet Power Spectrum of Varve Thickness**

[Figure]

**Wavelet Power Spectrum of log VT**

[Figure]

Fig. Wavelet analysis of the 1,500-varve record from Grand Lake, Labrador. White contours delimit areas of >95% confidence levels. Statistical significance was assessed using 100 Monte Carlo simulations that were performed to estimate the distribution of wavelet power under a red-noise (AR(1)) null model. The black dot represents a grid cell where the observed power exceeds the

significance threshold; they are helpful for visually spotting localized significant features within the broader white contour region. Short periods of around 7-year cycles were particularly active during 600-800AD, the Medieval climate anomaly (1100-1380) and the latter part of the Little Ice Age.

In terms of structure, the authors might consider moving Fig. 5 and the associated paragraphs to the discussion, after section 5.1. The attemps to determine the drivers will come more logically after having explained the depositional process of the varves.

> This is a great comment. We moved this part to the discussion (lines 273-283). Thanks

**Interpretations:**
Line 235: "One of the most notable features in the varve thickness series at GL is the major warm peak between the 1150s-1170s CE." I don't understand this sentence. In Fig. 5, the only predictor tested is precipitation, not temperature. The correlations with temperature proxies (not temperature records) only tell us that general trends in hydroclimates might be consistent regionally. Please rephrase accordingly.

> That is correct; however, paleo evidence suggests this period was likely characterized by elevated temperatures and higher sea surface temperatures (SSTs). This is supported by comparisons between our record and reconstructed Northern Hemisphere temperature datasets, including those reflecting AMV phases. These findings suggest that increased precipitation during this time was likely driven by warmer regional and/or hemispheric temperatures. Nevertheless, we rephrased the sentence accordingly (Lines 384-386).

As in most fluvial systems, the VT record might be affected by memory/autocorrelation effects (and nonlinear relationships between precipitation/flood strength/sediment transport). Did you try to use differentiation ($y_{diff}=y-y_{-1}$) to reduce these effects? Or investigate the autocorrelation patterns?

> We tested that by looking at the autocorrelation function (ACF) of the original and differenced varve thickness (VT) record over the past 1,500 years. The figure below was added as supplementary figure S5, and a sentence was added in lines 343-344.
>
> The VT time series shows strong autocorrelation at lags extending up to at least 60 years (left panel). When applying the same analysis to the differenced series (right panel: $Y_t = X_t - X_{t-1}$), the data appear more random, with values falling within the dashed confidence bounds. This indicates that the original autocorrelation likely stems from a long-term trend rather than true cyclicity. There is strong evidence from paleoclimate records suggesting this trend may be linked to long-term oceanic variability, such as the Atlantic Multidecadal Variability (AMV). Additionally, spectral analysis reveals significant multidecadal variability, with notable peaks around 48 and 76 years (which could be linked to AMV).

[Figure]

Fig. S5. Autocorrelation functions (ACF) of the original and differenced varve thickness (VT) record from Grand Lake over the past 1,500 years. The ACF of the original VT series (left panel) shows positive autocorrelation at short lags, with values remaining above the 95% confidence bounds at least until 60 years lag. This indicates the presence of low-frequency persistence, likely related to long-term climate variability. In contrast, the ACF of the first-differenced VT series (right panel) displays values that mostly fall within the 95% confidence bounds, consistent with white noise. This suggests that the autocorrelation observed in the original series is primarily driven by long-term trends rather than cyclic behaviour.

I found the discussion on weather patterns interesting and have no comments on that part.

Thank you.

A purely curiosity-driven question: do the authors notice changes in interannual variability of VT (potentially precipitation) throughout the record?

Detecting changes in year-to-year variability is challenging due to the high level of noise in the varve thickness record. We think that the NAO plays a significant role, as evidenced by the marked reduction in precipitation during the 1990s, which coincided with a pronounced intensification of the NAO+. We are currently preparing a second paper that applies change point analysis to identify and characterize abrupt shifts in the time series. This work also aims to examine how these shifts align in timing and phase with other high-resolution proxy records from the subpolar gyre.

**Figures:**
Figure 2: pannel b is hard to see in an otherwise well-constructed and clear figure. Pannel c: what do the codes mean (abp, abo, ab…?)

We added an explanation in the figure caption. Thanks

In Fig. 5,6,7: please add underlines or boxes to show where the MCA and the LIA are.
Figure 6: please plot the original data with the resampled ones so that one can keep track of the data transformation.

We added the MCA and the LIA limits in figures 4, 7 and 8. For Figure 7 (previously 6), we do not consider it necessary to plot the original data again, as it is already presented in Figure 4. For comparison with other proxy records at their lowest temporal resolution, we simply resampled the data by averaging over 10- or 4-year intervals.

**Technical comment**: There is ample litterature to emphasize the need to use a log-ratio transformation of xrf data (which are by essence compositional and therefore constrained, see for instance *Kucera and Malmgren, Marine Micropaleontology 1998*). Data presented as count rates can provide spurious results and should therefore be transformed (see recent publication by *Bertrand et al., Earth Science Reviews 2024*, with many clear examples).
I would therefore recommend that the authors revise figure 2 to show the iron relative concentration as log ratios (or centred log-ratios), with a clear x-axis and unit given.

We are perfectly aware that central log-ratios (clr) are the best way to use the XRF compositional datasets. However, clr are needed to help interpret variations of a group of elements over an entire sediment sequence and to deal with potential biases due to, for instance, compaction. Here, we only use Fe peaks to help identify the clay caps of the varves over a very short interval. In this very case, we don't think that log-ratios will provide spurious results. We are therefore respectfully declining the invitation to revise figure 2.

**Minor mistakes:**
Line 156: Three years *were* added
Thanks, changed.

*EDITOR COMMENTS:*

**1- Teleconnection**: make sure that the mechanism linking the NAO to your record is clearly spelled out in the main text (not only in the Suppl. Info), including a discussion on the relationship between NAO and winter precipitation. A justification of the uniqueness of your record (vs. why other records - which rely on summer reconstructions - might have not captured the signal) would also benefit the readers.

We added 21 lines in section 5.5 to follow this suggestion (previously 5.4 Teleconnection influencing temperature and precipitation).

2- Please make sure to integrate your replies to both reviewers' comments in the next version of the document (in particular, the methodological and statistical comments made by reviewer 2). I found your answers to be suitable and effective. Please include the updated periodogram and wavelet analysis figures to the manuscript

We included the recommended changes about the methodology and the statistical comments. The periodogram and the wavelet analysis are now in Figure 5 of the main text.

**References**

Nasri B.R., Boucher E., Perreault L., Remillard B.N., Huard D., Nicault A. & Projects A.-P. (2020). Modeling Hydrological Inflow Persistence Using Paleoclimate Reconstructions on the Quebec-Labrador (Canada) Peninsula. *Water Resources Research* 56, 5: e2019WR025122.